# Evaluation of Gut Microbiota in Healthy Persons and Type 1 Diabetes Mellitus Patients in North-Western Russia

**DOI:** 10.3390/microorganisms11071813

**Published:** 2023-07-15

**Authors:** Alexei B. Chukhlovin, Vasilisa V. Dudurich, Aleksey V. Kusakin, Dmitry E. Polev, Ekaterina D. Ermachenko, Mikhail V. Aseev, Yuri A. Zakharov, Yuri A. Eismont, Lavrentii G. Danilov, Oleg S. Glotov

**Affiliations:** 1R.M.Gorbacheva Memorial Institute of Oncology, Hematology and Transplantation, Pavlov First Saint Petersburg State Medical University, 197022 St. Petersburg, Russia; kusakin@scamt-itmo.ru (A.V.K.); y-eis@inbox.ru (Y.A.E.); 2Pediatric Research and Clinical Center for Infectious Diseases, 197022 St. Petersburg, Russia; olglotov@mail.ru; 3Serbalab Laboratory, 199106 St. Petersburg, Russia; vasilisadudurich@yandex.ru (V.V.D.); brantoza@gmail.com (D.E.P.); yermachenko@cerbalab.ru (E.D.E.); micas@mail.ru (M.V.A.); 4SCAMT Institute, ITMO University, 191002 St. Petersburg, Russia; 5Canadian Institute for Regenerative Medicine, Toronto, ON M5G OA3, Canada; prof.zakharov@gmail.com; 6Department of Genetics and Biotechnology, Saint-Petersburg State University, Universitetskaya Nab. 7/9, 199034 St. Petersburg, Russia; lavrentydanilov@gmail.com

**Keywords:** microbiota, bacterial, biodiversity, stool, 16S rRNA sequencing, diabetes mellitus, *Bacillota*, *Faecalibacteria*, *Ruminococcae*

## Abstract

Bacterial microbiota in stool may vary over a wide range, depending on age, nutrition, etc. The purpose of our work was to discriminate phyla and genera of intestinal bacteria and their biodiversity within a healthy population (North-Western Russia) compared to the patients with type 1 diabetes mellitus (T1DM). The study group included 183 healthy persons 2 to 53 years old (a mean of 26.5±1.0 years old), and 41 T1DM patients (mean age 18.2±1.8 years old). The disease onset was at 11±1.5 years, with a T1DM experience of 7±1.5 years. Total DNA was isolated from the stool samples, and sequencing libraries were prepared by amplifying the V3–V4 region of the 16S rRNA gene sequenced by Illumina MiSeq. Bioinformatic processing of NGS databases was adapted for microbiota evalutaion. Despite the broad scatter, the biological diversity for bacterial microbiota expressed as the Shannon index was significantly increased from younger to older ages in the comparison group, higher in adult healthy persons, with a trend for decrease in the *Actinomycetota* phylum which includes *Bifidobacterium longum* species. Similar but non-significant age trends were noted in the T1DM group. Concordant with the *Bacillota* prevalence in stool samples of diabetic patients, some anaerobic bacteria (*Faecalibacteria*, *Lachnospira* and *Ruminococcae*, *Roseburia*) were enriched in the T1DM microbiome against controls. Hence, correction of microbiota for *Ruminococcus* and *Lachnospiraceae* requires future search for new probiotics. Lower abundance of *Actinomycetota* and *Bifidobacter* in T1DM suggests potential usage of Bifidobacter-based probiotics in this cohort.

## 1. Introduction

Over the last decade, high diversity of intestinal bacterial microbiome was shown by means of new-generation DNA sequencing (NGS) techniques which enabled to detect hundreds of poorly cultured bacterial species in stool samples. In the general population, even the ratios of major bacterial classes in stool may vary within a broad range, depending on numerous factors, e.g., age, gender, immunity, nutrition style, ethnicity, etc. DNA isolation techniques may also sufficiently alter the results of NGS for the major bacterial phyla [1]. Hence, there is no single opinion on the normal ranges of intestinal microbiota acceptable for widespread clinical usage [2]. However, detectable changes in intestinal bacteriome (e.g., reduced *Firmicutes*) are shown to be associated with inflammatory bowel disease (IBD) accompanied by altered production of fatty acids and other metabolites by gut bacteria [3]. An excess of *Bacteroides* and a lower biodiversity of microbiota are observed in aged persons, especially in those with age-related chronic conditions and neurodegenerative diseases [4].

Potential reasons for impaired intestinal microbiota are also intensively studied in various metabolic disorders, in particular in type 1 diabetes mellitus (T1DM). This chronic insulin-dependent condition is caused by a deficiency in insulin production due to autoimmune damage to pancreatic islet cells [5]. The incidence of T1DM in the general population is about 1/200; most often, it manifests itself in childhood. Early detection of T1DM is based on the presence of specific autoantibodies and increased levels of glycosylated hemoglobin A1c (HbA1c). Clinical symptoms in diabetes include hyperglycemia, polydipsia, polyuria etc. The clinical symptoms of T1DM seem to be associated with the composition of the intestinal microbiota [6].

In view of the variable pattern of normal gut microbiota, several meta-analyses yielded a lot of controversial data on altered gut microbiota in T1DM when studying different populations from several parts of the world. These results are quite difficult to compare due to the differences in genetic background, diet and environmental factors [7,8,9]. For example, Siljander et al. [7] discussed the possible role of these factors for autoimmune diabetes in pediatric and young patients. In particular, they summarized a large number of studies on stool microbiota in T1DM where the authors showed some relations with various genera and species of *Firmicutes* (*Bacillota*), *Bacteroides*, *Proteobacteria*, etc. Some characteristic alterations of intestinal microbiota may play a role in the autoimmune genesis of the disease [10].

Therefore, the purpose of our work was to assess predominant phyla and genera of intestinal bacteria and their biodiversity among the healthy population of the North-West of Russia, taking into account the basic demographic factors, i.e., the age and gender of the examined individuals. We also studied gut microbiota in the patients with type 1 diabetes mellitus, a metabolic autoimmune disorder with decreased insulin production. The regional reference values for biodiversity and main types of normal intestinal microbiota were determined, and a number of significant changes were revealed between normal persons and patients with T1DM.

## 2. Materials and Methods

### 2.1. Ethical Statement

All methods of collecting biological samples from stool used under this medical examination were taken only with the approval of the attending physician. The study was conducted in accordance with guidelines of the 1964 Declaration of Helsinki and its later amendments. All patients or their guardians signed a written informed consent form for a hematopoietic stem cell transplant and the subsequent medical procedures, as well as potential usage of their clinical data for the purposes of clinical research. This study was approved by the Local Review Board of the Pavlov First State Medical University of St. Petersburg (ID number 214 of 17 December 2018).

### 2.2. Patients and Controls

Our population study was performed from January 2021 to December 2022. The study group included 41 patients with type 1 diabetes mellitus (T1DM) aged from 5 to 48 years (mean age 18.2±1.8 years old) (Table 1). The female-to-male ratio was 42% and 58%, respectively (17/24). The mean age of T1DM onset was 11±1.5 years, with the disease experience of 7±1.5 years. A total of 27% of the patients (n = 11) were under 10 years old. The control group included 183 healthy persons aged 2 to 53 years old, with the mean age of 26.5±1.0 years old. The female-to-male ratio in controls was 62% to 38% (113/70 cases). The studied group of T1DM patients was selected by the conventional features of clinically confirmed T1DM. The exclusion criteria were as follows: absence of other endocrine, gastrointestinal and hepatic diseases, beyond exacerbation of autoimmune disorders, viral or bacterial infections (including COVID-19) within the last 6 months; absence of treatment with antidepressant or neuroleptic drugs, as well as statins and proton pump inhibitors within the last year. The control group was recruited from the healthy population, in absence of inflammatory bowel diseases and type 2 diabetes mellitus, without oncological diseases, bacterial or viral infections (including COVID-19), and antibiotic usage within the last 6 months, as well as non-application of certain drugs (see above) within the past 12 months.

### 2.3. Sampling, DNA Isolation and Preparation of the Sequencing Libraries

Collection of fecal samples was performed at home. The specimens were taken from toilet paper, followed by sampling of the inner substance and immediate transfer to a sterile Eppendorf-type tube with 0.5 mL of an EDTA-based transport medium. The average mass of samples was 0.1 m, not exceeding 0.5 mL. The sample was immersed in the medium with a mucolytic substance (InterLabservice, Moscow, Russia). Transportation of the biological samples took 3 to 12 h at 4 °C followed by a DNA isolation procedure. Freezing of the material during storage and transportation was avoided.

Total DNA was isolated from the suspended stool samples subjected to homogenization in a lysing solution homogenized with the bead technique, followed by DNA extraction by the sorbent column technique (Qiagen, Germantown, MD, USA) according to the manufacturer’s recommendations.

The 16S DNA sequencing libraries were prepared according to Illumina’s 16S Metagenomic Sequencing Library Preparation protocol (Part #15044223 Rev. B). The reagent kits were purchased from Illumina (Illumina, San Diego, CA, USA). We used 5 ng of total DNA per sample in order to amplify the target fragment of the 16S rRNA gene by means of the recommended primers for the V3–V4 region. We performed 25 PCR cycles using the KAPA HiFi HotStart ReadyMix (2×) (Roche Diagnostics, Zug, Switzerland). After purification of PCR products with the SPRI bins, we indexed 5 ng of the resulting amplicons with the KAPA HiFi HotStart ReadyMix (2×) (Roche Diagnostics, Zug, Switzerland) and the Nextera XT Index Kit (Illumina, San Diego, CA, USA). We ran 8 cycles of index PCR according to the Illumina protocol. The obtained libraries were sequenced using the Illumina MiSeq platform.

### 2.4. Statistics and NGS Data Processing

Bioinformatic processing of the 16S DNA database was carried out using a custom bioinformatics pipeline implemented in the programming languages R v.3.6 and Python3. At the first stage of the pipeline, the primer sequences were truncated at the beginning of paired reads, while the pairs of reads that did not contain primer sequences were discarded. Further on, we trimmed 25 base pairs from the end of each read (low-quality bases) and processed the resulting data using the DADA2 pipeline to identify exact sequence variants [11]. After defining exact sequence variants, the forward and reverse reads were concatenated and the resulting sequences were used for Naive Bayes taxonomic classification [12] using the SILVA v138 reference database [13]. Identification of bacterial species was performed using the exact matching algorithm in DADA2 using SILVA v138 sequences pre-processed in an appropriate way using the custom scripts.

To evaluate the relative contents of bacterial phyla, genera, and species in control and T1DM groups, we used standard STATISTICA 5.0 software. The descriptive results for distinct groups were expressed as M±m, median, minimal and maximal values. The intergroup differences were evaluated by parametric methods (*t*-test) or a non-parametric U test. Correlation quotients were assessed by means of a non-parametric Spearman criterion. The confidence levels of *p* < 0.01 were considered statistically significant. Statistical differences in abundance of bacterial taxa for different cohorts were assessed by means of non-parametric Mann–Whitney U test for paired comparisons. For multiple testing, the correction was made using the Benjamin–Hochberg method in R. To calculate the Shannon Diversity Index, a matrix containing total numbers of amplicon sequence variants (ASVs) at the species level per sample was provided as the input to the “vegan” package in the R programming language [14]. In general, upon bioinformatic processing of databases, we used a generally accepted set of statistical tools adopted for calculation of the microbiota components [15]. To identify special taxa for each group, sparse partial least squares discriminant analysis (sPLS-DA) was carried out with the “multiomix” software (mixOmics software package) in the R programming language [16]. In this case, discriminant analysis of samples was performed in order to detect the parameters which maximize the differences between the compared groups [17]. This method is suitable for visual presentations of results in microbiota studies [15,18].

## 3. Results

### 3.1. Age-Dependent Changes in Microbiota in Controls

The control group included 39 children at the age of 3 to 10 (Group 1), 21 adolescents and young adults (11 to 21 years old) and 123 adult persons (Group 3, 22 to 53 years old). NGS analysis included the results of 16S rRNA genotyping in 183 stool samples from these individuals who live in the North-Western Russia (mostly in St. Petersburg). Biological diversity of gut microbiota for different age groups in the control population was assessed via the Shannon index (Figure 1). Despite a broad scatter of individual results, we revealed a significant increase in mean biodiversity index in the second group (adolescents) compared to younger children (first group) (W = 246, *p*-value = 0.015 by U test). This difference was even more pronounced for adult persons (Group 3; W = 1475, *p*-value = 0.00015). Hence, one may conclude on the bacterial species enrichment from childhood to adult microbiota.

### 3.2. Phyla and Genera

The detection frequencies of specific types (Phyla) and genera (Genera) of intestinal bacteria in this populational sample are shown in Table 2. The bulk of normal intestinal microorganisms consisted of *Firmicutes* (*Bacillota*) and *Bacteroidota* (a total of 88%) followed by the much less presented *Proteobacteriota* and *Actinomycetota*, thus corresponding to the generally accepted normal ratios for human gut microbiota. At the same time, the minimal and maximal values, especially for rare phyla, fluctuate over a very wide range (see Table 2). It should be noted that unlike average values, the zero median values are often seen for rare genera and species of bacteria, meaning the detection of their specific DNA sequences in less than 50% of the samples. For example, in Table 2, this applies to *Campylobacterota*, *Fusobacteriota*, *Euriarcheota*, and *Synergistota*. Table 2 shows that there is a weak age-dependent decrease for the *Actinobacteriota* type. Meanwhile, no significant age dependence was found for other dominant bacterial types (*Firmicutes*, *Bacteroidota*).

Table 3 presents the relative incidence of intestinal bacteria by their genera. At this level of taxonomy, significant age-dependent changes are seen for several genera: *Phascolarctobacteria*, *Parabacteroides*, *Bifidobacteria*, *Coprococcus*, *Dorea*, *Haemophylus*, *Coprobacter*, *Veillonella*, *Flavonifractor*.

### 3.3. Correlation between Bacterial Genera and Species

We also attempted to compare the age-dependent trends of intestinal microbiota in controls for distinct bacterial species. When analyzing different bacterial genera and appropriate species, we found that similar age-dependent trends (either positive or negative correlations) are shown for appropriate bacterial species (Table 4). Hence, the data from NGS sequencing allow the detection of sufficient age-dependent trends in microbiota at the level of genera in the complex bacterial mixtures.

### 3.4. Differential Features of Gut Microbiota in Type 1 Diabetes Mellitus

Due to sufficient age-dependent changes in normal microbiota, we compared the Shannon diversity index for the three age groups in T1DM patients aged 5 to 48 years (Figure 2). The distribution by age group was the same as for the control group, i.e., Group 1 included patients under 10 years of age; Group 2 included patients 11 to 21 years of age; Group 3 included patients over 22 years of age. This parameter tends to decrease in adult patients; however, it does so without significant differences due to the small size of the sample and the high scatter of individual data (Wilcoxon test with Bonferroni corrections, with *p* = 0.76 for Groups 1–2; *p* = 0.67 for Groups 1–3; *p* = 0.76 for Groups 2–3).

The Shannon diversity index in the total groups of patients and the controls is presented in Figure 3. Indeed, this parameter is lower in the T1DM group. The changes did not, however, reach statistical significance (W = 2646, *p*-value = 0.209).

### 3.5. Differences between T1DM and Controls for the Main Taxonomic Units

When evaluating the data on bacterial phyla, we compared their incidence in samples from controls and T1DM cases (Table 5). The prevalence of major phyla (*Firmicutes*/*Bacillota* and *Bacterioidota*) proved to be within 40–50%, thus making up about 90% of entire bacterial microbiota. Other phyla were presented at much lower rates. A low number of DNA sequences (up to 0.3%) could not be classified. Of interest, a higher prevalence of *Firmicutes* (*p* = 0.0009), along with a lower presentation of *Actinomycetota* (*p* = 0.01), may be noted in the stool samples from diabetic patients.

In general, an obvious reverse correlation is revealed between *Bacteroides* and *Firmicutes* in the total group of fecal samples (Figure 4). This interrelation may be connected with their different metabolic roles in healthy and diabetic gut microbiota.

Moreover, upon evaluation of the Firmicutes-to-Bacteroides ratio, a standard microbiota index, we revealed its significant increase in T1DM patients compared the control group as shown in Figure 5.

Evaluation of gut microbiota by genera was also performed. Appropriate data for the most common genera are presented in Table 6. Meanwhile, the more rare genera (<0.5%) are detected in the vast minority of samples, and therefore they were not subject to comparison in population studies, both control and diabetes patients. Of special interest, several anaerobic Gram-positive bacteria (*Faecalibacteria*, *Lachnospira* and *Ruminococcae*, *Roseburia*) proved to be more common in the diabetic microbiome than in controls, thus being in compliance with the higher ratio of *Firmicutes* in the patients. Low relative content was shown for some clinically actual microorganisms, e.g., *Escherichia*, *Streptococcus* without any differences between controls and T1DM. Meanwhile, the unclassified DNA sequences constituted a sufficient fraction of the total sequences (a mean of 13–14%), thus causing their underestimation in total data analysis.

### 3.6. Species Differences between T1DM Patients and Control Group

Table 7 contains the data on the relatively common bacterial species which showed differences between controls and T1DM. Most bacterial species increased in T1DM belong to *Firmicutes* (*F. prausnitzii*, *Blautia* spp., *Veillonella dispar*). Moreover, we revealed a decrease in *Bifidobacteria* spp., *Collinsella aerofaciens* which belong to the *Actinomycetota* phylum.

## 4. Discussion

We studied a representative control group (183 persons) in order to assess the phylum-specific reference ranges as well as the *Firmicutes*/*Bacteroidetes* ratio for the healthy population. Limitations of the present study include a relatively small number of petients in the T1DM group. These results should be confirmed in further clinical series.

The Shannon diversity index was used to evaluate the diversity of bacterial microbiota. When studying age dependence of gut microbiota, we found a trend for a decrease in the *Actinobacteriota* phylum. Interestingly, this phylum includes the *Bifidobacter* species, which are of great metabolic significance and prevail in the infant microbiota [19,20]. *Actinobacteriota* are exhausted in older age groups, as shown by our results. This finding suggests a potential benefit of the usage of Bifidobacter-based probiotics in these patients.

Variable frequency of major bacterial phyla (*Firmicutes* and *Bacteroidetes*) was registered in all subgroups of normal persons and T1DM patients. Reduced microbial diversity in diabetic patients is a common finding, as reviewed by Zhou et al. [21]. However, the Shannon index of diversity in our study did not show significant differences between T1DM and the control group.

Comparing the bacterial composition of the microbiota in patients with diabetes mellitus and in the control group by types and genera, the most pronounced feature is a slight but significant predominance of *Firmicutes* over *Bacteroides* in the group of patients with T1DM. This difference is consistent with the results obtained by Pellegrini et al. [22], who showed an increase in *Firmicutes* and the *Firmicutes*/*Bacteroidetes* ratio in the duodenal microbiome in diabetes mellitus, which correlated with the increased gene expression of several pro-inflammatory chemo- and cytokines in the intestinal mucosa. In general, however, the baseline estimates for the *Firmicutes*/*Bacteroidetes* ratio in children with T1DM show conflicting results as seen from the published meta-analyses [8,23]. A more detailed analysis on the levels of bacterial genera showed an increased content of certain *Firmicutes* in T1DM patients, in particular *Faecalibacteria*, *Ruminococci*, *Lachnobacteria*, and *Roseburia*, thus confirming the predominance of the *Firmicuta* (*Bacillota*) phylum in diabetes patients.

An in-depth study by van Heck et al. [24] reported on an increased content of *Ruminococcaceae* and *Clostridiaceae* families in the faeces of patients with type 1 diabetes (240 cases). The members of *Ruminococcaceae* are very sensitive to antibiotic therapy. For example, lower gut levels of *Ruminococcaceae* have been consistently observed from baseline to day 7 of diarrhea caused by antibiotic therapy, i.e., with amoxicillin–clavulanate [25]. The authors suggest that a decrease in *F. prausnitzii*, a member of the *Ruminococcus* genus, may predict the risk of diarrhea due to antibiotic treatment.

Another microorganism that is less abundant in T1DM, *Collinsella aerofaciens*, belongs to *Actinomycetota*, such as *Bifidobacter*, thus consistent with the general reduction in this bacterial phylum in our group of diabetic patients. There are only scarce data on the physiological significance of *Collinsella*. For example, enriched *Collinsella aerofaciens* and some *Dorea* species in stool were shown to be associated with obesity [26].

We also found a higher frequency of *Veillonella dispar* in patients with type 1 diabetes. These anaerobic bacteria belong to *Firmicutes* (*Bacillota*), which are more common in our diabetic group. This finding may be also interpreted in terms of protective effects against certain enteric pathogens, such as *C. difficile* infection [27]. The authors also suggested that the baseline microbiota spectrum before antibiotic treatment could predict *C. difficile* recurrence. This conclusion still needs further confirmation. From a metabolic point of view, an increased content of *Veillonella dispar* may be considered a sign of chronic hypoxia. A relative increase in these bacteria has been found in high-intensity athletes, probably due to lactate hyperproduction [28]. *Roseburia* levels have also been shown to decrease in patients with type 1 diabetes. These bacteria produce short-chain fatty acids which exert various anti-inflammatory immune effects. Again, they belong to the phylum *Firmicutes*, *Lachnospiraceae* family, as reviewed by Nie et al. [29].

*Roseburia intestinalis* and *F. prausnitzii* are among the most active producers of butyrate in the intestinal microbiota [30]. Meanwhile, butyrate supplementation in the diet leads to better control of hyperglycemia in patients with type 1 diabetes [31]. Therefore, the abundance of butyrate-producing bacteria may be beneficial for diabetic patients.

A number of reviews consider altered representation of bacterial genera in T1DM, thus probably reflecting the potential effects of genetic factors, diet, and autoimmune background in these patients. In particular, several studies of microbiota in T1DM suggest some relationships with various genera and species of *Firmicutes*, *Bacteroides*, *Proteobacteria*, etc. Of interest is the work by Spanish authors [32] which concerned children with type 1 diabetes. The disorder was associated with a decreased diversity of the microbiota. The relative predominance of *Ruminococcus*, *Blautia* (related to *Firmicutes*), and *Veillonella* was combined with a reduced content of *Bifidobacterium*, *Roseburia*, *Faecalibacterium* and *Lachnospira*. In general, however, the studies of gut microbiota in type 1 DM conducted in different countries and populations yielded rather controversial results due to the diversity of genetic backgrounds, diet and environmental factors, as evidenced by the available meta-analyses [7].

In particular, a special study concerned the gut microbiota composition in Chinese patients with diabetes and that of the control group [33]. As in other works, the dominance of some *Bacteroidetes* and *Firmicutes* was shown. Of note, an excess of *Faecalibacterium* showed an inverse correlation with glycosylated hemoglobin (HbA1c) levels, thus probably depending on the stage of disease and glycemia control in individual patients. A similar correlation was also found between the contents of *Ruminococcaceae* and the levels of autoantibodies to pancreatic islet cells.

Therefore, a potential correction of microbiota, especially *Ruminococcus* and *Lachnospiraceae*, could be the subject of future search for new probiotics. For example, *Faecalibacterium prausnitzii*, belonging to *Ruminococcus*, is a producer of butyrate, which plays an important role in diabetes control. Therefore, the transplantation of biological preparations containing *F. prausnitzii* may be chosen as a strategy for the treatment of intestinal dysbiosis associated with inflammation, which contributes to the development of autoimmune disease and diabetes. Some authors consider *F. prausnitzii* a potential probiotic, especially in protecting the gut microbiota and its therapeutic potential against inflammation and diabetes [34]. In particular, a lower abundance of *Actinomycetota* and *Bifidobacter* in T1DM group suggests a potential usage of Bifidobacter-based probiotics in this cohort.

## 5. Conclusions

The general feature of gut microbiota in T1DM patients is a significant increase in the members of *Firmicutes* phylum. This finding may be caused by mechanistic reasons (genetic background, autoimmunity, insulin dependence, hyperglycemia) and environmental factors, e.g., limitations in diabetic diet and lifestyle. The study has some limitations, e.g., a relatively small number of diabetic patients. An extended analysis of the regional population, especially among the pediatric cohort, is required later. In the future, it is necessary to perform a detailed analysis of the associations between clinical factors, immunological parameters, and the composition of the intestinal microbiota determined by standardized NGS methods in distinct regional populations [9]. The results obtained can be applied in the future to find new ways to correct the microbiota by means of well-known and new-generation probiotics.

## Figures and Tables

**Figure 1 microorganisms-11-01813-f001:**
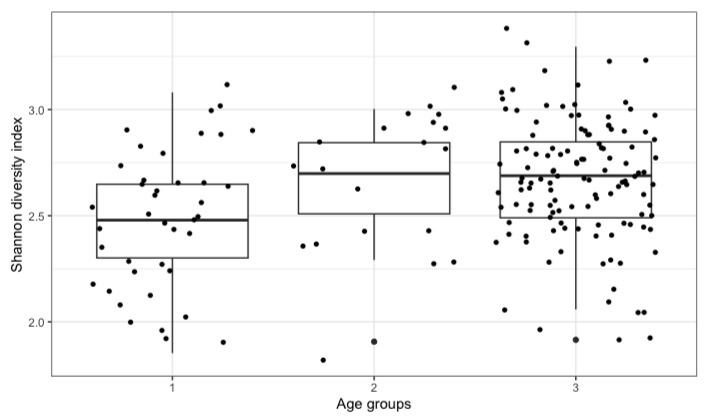
Biodiversity index of fecal bacterial species in the control group (n = 183). The age groups 1 to 3 are specified above (Section 3.1). Abscissa: age group; Ordinate: individual values of the Shannon index.

**Figure 2 microorganisms-11-01813-f002:**
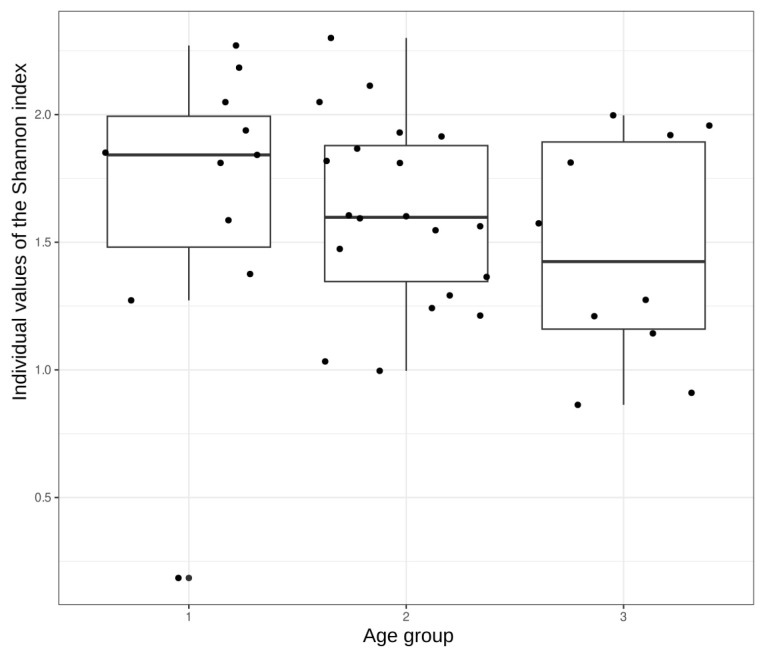
Biodiversity index of fecal bacterial species in the diabetes group (n = 41). The age groups 1 to 3 are specified above (Section 3.1). Abscissa: age group; Ordinate: individual values of the Shannon index.

**Figure 3 microorganisms-11-01813-f003:**
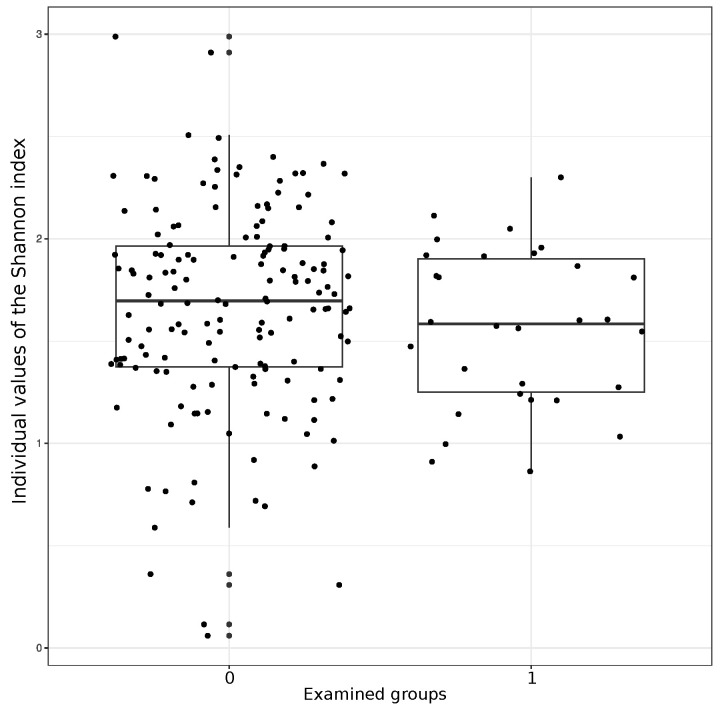
Individual values of the Shannon biodiversity index for intestinal microbiota in control group (left) and T1DM patients (right graph). The age groups 1 to 3 are specified above (Section 3.1). Abscissa: examined groups (0, controls; 1, T1DM); Ordinate: mediane and individual values of Shannon index.

**Figure 4 microorganisms-11-01813-f004:**
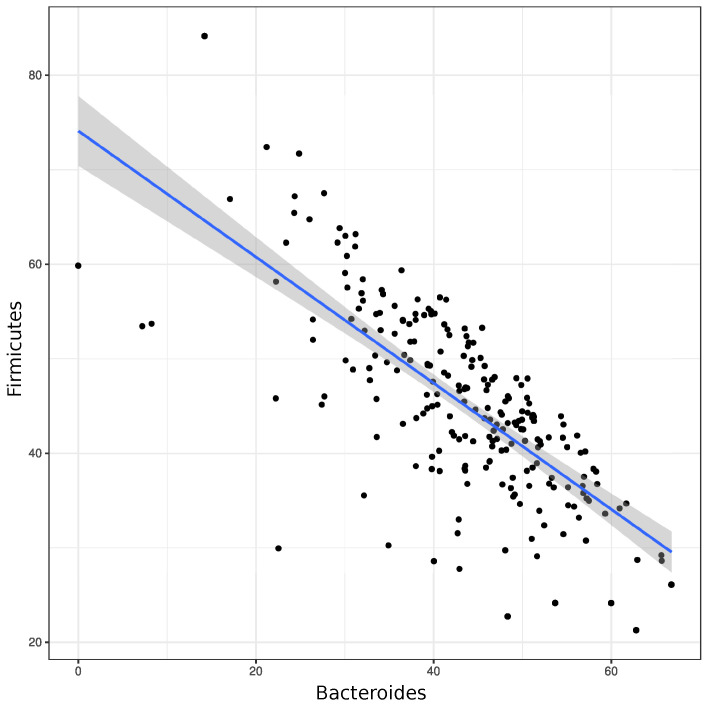
Reverse correlation curve between relative contents per cent of total bacterial mass of *Bacteroides* and *Firmicutes* in the total group of samples (n = 224; r quotient = −0.7379265, S = 3,255,484, *p*-value < 2.2 × 10${}^{-16}$).

**Figure 5 microorganisms-11-01813-f005:**
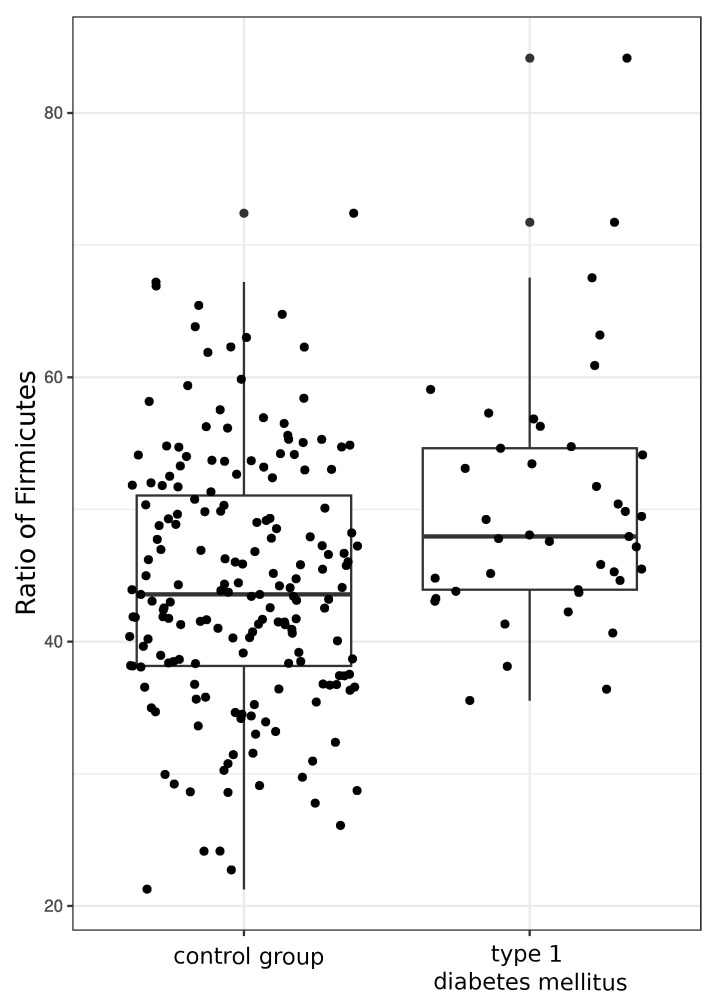
Ratio of *Firmicutes* to the total bacterial mass in gut microbiota of control group (left) and type 1 diabetes mellitus (right). The median level of this parameter is increased in T1DM (Wilcoxon test, W = 2506; *p*-value = 0.0009).

**Table 1 microorganisms-11-01813-t001:** Demographic data of T1DM patients and comparison (control) group.

Demographic Parameters	Diabetes Mellitus Patients (n = 41)	Comparison Group (n = 183)	*p*-Value
Mean age (M + m), years old	18.2±1.8	26.5±1.0	$3.069\times 10^{-5}$
Median age (min–max values)	18.2 (5–48)	26.7 (3–53)	0.000146
Age groups (years old):	Number of cases (per cent of total)		
1. 3–10 2. 11–21 3. >22;	12 (29%) 16 (39%) 13 (32%)	39 (21%) 21 (12%) 123 (67%)	0.8109 0.7761 0.0876

The Wilcoxon test was used for comparing median values and *t*-test was used for mean comparing.

**Table 2 microorganisms-11-01813-t002:** Relative contents (% of the total number of sequences, frequency > 0.5%) of the most common bacterial phyla in the control population of the North-Western Russia (n = 183).

Phyla	Detection Rate in the Sample, % of General Presentation M±m/Min–Max Values	Correlation with Age, r Quotient (Spearman Criterion)	Confidence *p*-Value
*Firmicutes* (*Bacillota*)	44.20±0.73 7.0–72.4 (43.6)	0.095	0.101
*Bacteroidota*	43.51±0.79 0–66.8 (43.8)	−0.008	0.457
*Proteobacteriota*	3.96±0.26 0.03–29.6 (3.06)	0.076	0.155
*Actinomycetota*	3.78±0.33 0–23.78 (1.96)	−0.185	**0.006**
*Verrucomicrobiota*	2.83±0.42 0–44.1 (0.38)	−0.037	0.309
*Desulfobacteriota*	0.50±0.07 0–9.93 (0.29)	0.129	0.041

**Table 3 microorganisms-11-01813-t003:** Relative contents (% of the total number of sequences, frequency > 0.5%) of the most common bacterial genera in control population of the North-Western Russia (n = 183).

Genera	Detection Rate in the Sample, % of General Presentation M±m/Min–Max Values	Correlation with Age, r Quotient (Spearman Criterion)	Confidence *p*-Value	Phyla/Families
*Bacteroides*	24.70±1.00 0–56.1 (24.6)	−0.150	**0.020**	*Bacteroidota*
*Prevotella*	7.11±0.86 0–50.70 (0.23)	0.157	**0.017**	*Bacteroidota*
*Faecalibacteria*	6.25±0.27 0–17.54 (5.89)	−0.051	0.245	*Bacillota*
*Alistipes*	3.13±0.21 0–13.28 (2.47)	0.036	0.312	*Bacteroidota*
*Akkermansia*	2.51±0.40 0–44.09 (0.11)	−0.036	0.312	*Bacillota*
*Phascolarctobacteria*	2.31±0.21 0–12.78 (1.21)	0.262	**0.0002**	*Bacillota*/Negativicutes
*Parabacteroides*	2.21±0.16 0–9.45 (1.67)	0.248	**0.0004**	*Bacteroidota*
*Lachnospira*	2.23±0.25 0–22.89 (1.18)	−0.021	0.389	*Bacillota*
*Bifidobacterium*	2.10±0.24 0–17.20 (0.77)	−0.254	**0.0003**	*Actinomycetota*
*Agathobacter*	1.76±0.18 0–15.66 (0.97)	0.029	0.349	*Bacillota*
*Dialister*	1.65±0.18 0–17.08 (0.29)	−0.165	**0.010**	*Bacillota*/*Veillonales*
*Ruminococcus*	1.63±0.18 0–14.88 (0.67)	0.060	0.208	*Bacillota*
*Roseburia*	1.24±0.11 0–10.06 (0.72)	0.00005	0.500	*Bacillota*
*Sutterella*	1.01±0.10 0–8.52 (0.53)	0.156	0.018	*Pseudomonadota*
*Collinsella*	0.95±0.16 0–21.18 (0.31)	0.019	0.401	*Actinomycetota*
*Parasutterella*	0.80±0.10 0–7.03 (0.11)	−0.078	0.147	*Pseudomonadota*/class *Betaproteobacteria*
*Coprococcus*	0.75±0.07 0–6.96 (0.44)	0.291	**0.00003**	*Bacillota*
*Erysipelotrix* UCG 003	0.71±0.09 0–7.51 (0.21)	0.163	**0.014**	*Bacillota*/class *Erysipelotrichia*
*Streptococcus*	0.69±0.17 0–26.69 (0.20)	−0.088	0.118	*Bacillota*/class *Bacilli*
*Blautia*	0.68±0.06 0–6.65 (0.45)	−0.033	0.327	*Bacillota*
*Escherichia*	0.66±0.12 0–13.57 (0.11)	0.064	0.194	*Pseudomonadota*/class *Proteobacteria*
*Paraprevotella*	0.65±0.10 0–11.68 (0.12)	0.169	**0.011**	*Bacteroidota*/class *Bacteroidia*
*Odoribacter*	0.59±0.05 0–3.31 (0.41)	0.027	0.360	*Bacteroidota*/class *Bacteroidia*
*Dorea*	0.44±0.04 0–3.30 (0.29)	0.264	**0.00015**	*Bacillota*/*Clostridia*

**Table 4 microorganisms-11-01813-t004:** Significant age-dependent trends at genera and species levels of intestinal bacteria in control group (n = 183).

Species	Detection Incidence, M±m/Min–Max Values	Correlation with Age, r Quotient (Spearman Criterion)	Confidence Levels P	Bacterial Phyla/Genera (Correlation with Age in Parentheses)
*Phascolarctobacterium faecium*	1.2±0.15 0–11.7 (0)	0.218	**0.002**	*Bacillota*/*Phascolarctobacteria* (r = 0.262)
*Faecalibacter prausnitzi*	1.13±0.10 0–9.0	−0.172	**0.010**	*Bacillota*/*Clostridia*
*Bifidobacterium longum*	0.47±0.08 0–10.4 (0.09)	−0.195	**0.004**	*Actinomycetota*/*Bifidobacterium* (0.254)
*Coprococcus comes*	0.18±0.0 0–2.0 (0)	0.276	**0.0001**	*Coprococcus* (0.291)
*Haemophilus influenzae*	0.14±0.06 0–9.0 (0)	−0.242	**0.0005**	*Haemophylus* (−0.214)
*Flavonifractor plautii*	0.10±0.02 0–2.2 (0)	−0.178	**0.008**	*Flavonifractor* (−0.200)
* **Veillonella dispar** *	0.024±0.008 0–3.3 (0)	−0.193	**0.004**	*Veillonella* (−0.215)

**Table 5 microorganisms-11-01813-t005:** Relative incidence of the main phyla in gut microbiota in control persons and T1DM patients.

Phyla	Control Group (n = 183)	Type 1 Diabetes Mellitus (n = 41)	*p*-Value
* **Firmicutes** * ** (** * **Bacillota** * **)**	44.4±0.7	50.2±1.5	**0.0009**
*Bacteroidota*	43.5±0.8	40.5±1.6	0.114
*Proteobacteriota*	4.0±0.3	4.4±0.7	0.553
* **Actinomycetota** *	3.8±0.3	2.8±0.9	**0.011**
*Verrucomicrobiota*	2.8±0.4	1.2±0.3	0.387
*Desulfobacteriota*	0.50±0.07	0.41±0.09	0.332
*Cyanobacteriota*	0.27±0.04	0.09±0.06	0.820
Unclassified	0.23±0.10	0.12±0.02	0.158
*Campilobacterota*	0.22±0.10	0.003±0.003	0.049
*Fusobacteriota*	0.166±0.117	0.006±0.005	0.593
*Euriarcheota*	0.10±0.03	0.17±0.07	0.238

Notes. 1. Incidence of different bacterial phyla is expressed as per cent of total amplicon sequences (M±m). 2. Bacterial phyla are placed by their frequency in samples (>0.5% of total presentation).

**Table 6 microorganisms-11-01813-t006:** Relative presentation of the main bacterial genera in control group and in T1DM patients (incidence rates >0.5% of total).

Genera	Control Group (n = 183)	Type 1 Diabetes Mellitus Patients (n = 41)	*p*-Value
*Bacteroides*	24.7±1.0	23.2±1.8	0.550
*Prevotella*	7.1±0.9	4.8±1.2	0.280
*Faecalibacteria*	6.3±0.3	8.9±0.8	**0.0005**
*Alistipes*	3.1±0.2	3.6±0.6	0.410
*Akkermansia*	2.51±0.40	1.14±1.31	0.792
*Phascolarctobacteria*	2.31±0.21	2.07±0.42	0.479
*Parabacteroides*	2.21±0.16	1.83±0.24	0.541
*Lachnospira*	2.2±0.3	3.1±0.4	**0.0004**
*Bifidobacter*	2.1±0.2	1.9±0.8	**0.003**
*Agatobacter*	1.76±0.18	1.95±0.37	0.768
*Dialister*	1.65±0.18	1.94±0.48	0.978
*Ruminococcus*	1.6±0.8	2.5±0.4	**0.005**
*Roseburia*	1.2±0.1	1.6±0.2	**0.032**
*Sutterella*	1.01±0.1	1.09±0.23	0.586
*Escherichia*	0.66±0.12	0.98±0.53	0.880
*Streptococcus*	0.69±0.17	0.49±0.12	0.756
Unclassified	13.2±0.5	14.4±1.2	0.373

**Table 7 microorganisms-11-01813-t007:** Selected species of intestinal bacteria which differed between controls and T1DM patients.

Bacterial Species	Control Group (n = 183)	Type 1 Diabetes Mellitus Patients (n = 41)	*p*-Value
*Faecalibacterium prausnitzii*	2.80±0.19	3.77±0.43	**0.030**
*Collinsella aerofaciens*	0.87±0.15	0.29±0.08	**0.008**
*Bacteroides caccae*	0.80±0.25	0.30±0.14	0.08
*Blautia obeum*	0.07±0.008	0.12±0.03	**0.01**
*Roseburia intestinalis*	0.18±0.06	0.08±0.05	**0.03**
*Veillonella dispar*	0.024±0.008	0.047±0.016	**0.003**

## Data Availability

We report no links to publicly archived datasets analyzed or generated during the study.

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
