# Peer review of "Evaluation of Gut Microbiota in Healthy Persons and Type 1 Diabetes Mellitus Patients in North-Western Russia"

_microorganisms, 2023, doi:10.3390/microorganisms11071813_

Round 1

Reviewer 1 Report

 Please the following points should be followed:

-          Abstract should be rewritten in more details and high lighting the main results in order to sound better and giving strength to the manuscript.

-          Abstract: write whole the words line 5, y. o. as years old.

-          Introduction was written with more details, especially with regard to healthy persons.

-          Key words: line 16, correct 16S rDNA to 16S rRNA.

-          Line 54: add ID no. of ethical approval.

-          Materials and methods:

·         Add missed references in all steps mentioned.

·         Line 62: please add a table for the mentioned data as source of collected samples as well as design and groups, with special reference to age, sex, …etc.

·         Line 65: check the % to totally be 100% not more.

·         Line 70: 113+69= 182 persons not 183 as mentioned in the manuscript, please check and correct.

-          Results:

·         Line 118 please add a table summarizing the mentioned result.

·         In the results of statistics, add letters a, b, c, …. etc to highlight the significance differences.

-          Discussion: should be rewritten with more details to cover all mentioned results, as well as comment with your suggestion about the increasing or decreasing of some genera with regard to different groups.

-          References: Should be updated till 2023.

The English quality is good but a minor editing is required please

Author Response

Much thanks for Your useful and constructive remarks

Our replies are as follows:

(1) Abstract should be rewritten in more details and high lighting the main results in order to sound better and giving strength to the manuscript.
Answer 1. Abstract is supplied with some additional data pointing to some specific bacteria revealed, as well as suggesting usage of Bifidobacter-besed probiotics in T1DM. Short abstract is in accordance with requirements of the Journal.

(2) Abstract: write whole the words line 5, y. o. as years old.
Answer 2. y.o. corrected throughout the text

(3) Introduction was written with more details, especially with regard to healthy persons.
Answer 3. We have added 2 additional references discussing high variability of biodiversity of microbiota in general population, both in health and disease (lines 27-33).

(4) Key words: line 16, correct 16S rDNA to 16S rRNA.
Answer 4. Done

(5) Line 54: add ID no. of ethical approval.
Answer 5. Ethical approval is mentioned with appropriate ID

Materials and methods:
(6) Add missed references in all steps mentioned.
Answer 6. In experimental section, we mentioned all the required technical references as well as statistical approaches with appropriate references.

(7)  Line 62: please add a table for the mentioned data as source of collected samples as well as design and groups, with special reference to age, sex, …etc.
Answer 7. We have added a table with details of age and gender parameters in T1DM and comparison group (Table 1, line 87)

(8) Line 65: check the % to totally be 100% not more.
Answer 8. Corrected

(9) Line 70: 113+69= 182 persons not 183 as mentioned in the manuscript, please check and correct.
Answer 9. Corrected

Results:
(10) Line 118 please add a table summarizing the mentioned result.
Answer 10. The scatterplot graph (Fig.1) seems reflect the distribution of data in diabetes and controls. An additional table could duplicate these graphical data.

(11) In the results of statistics, add letters a, b, c, …. etc to highlight the significance differences.
Answer 11. The age ranges for groups 1, 2 and 3 are provided in the legend to Fig.1. The levels of significance are shown above in the text (lines 124-127)

Discussion:
(12) Should be rewritten with more details to cover all mentioned results, as well as comment with your suggestion about the increasing or decreasing of some genera with regard to different groups.
Answer 12. Sufficient role of Bifidobacter is stressed in more details, and 2 appropriate references are provided under Discussion.

References:
(13) Should be updated till 2023.
Answer 13. Four quite recent works are referred in the Introduction and Discussion.

Reviewer 2 Report

The authors describe in interesting comparison of intestinal microbiota profiles in healthy people and DM type 1 patients. The findings important to understanding relationships between the microbiota and health/disease states.

Comments:

The language needs editing. For example, these and other sentences/words need correction:

Line 1. "sufficiently"

Line 20. "allowed to detect"

Abstract: the concluding sentence about new probiotics seems to have come out of no where. Perhaps conclude with the comparison of health to DM I microbiota populations.

Methods: Include a statement about IRB approval of the protocol.

List inclusion and exclusion criteria and how participants were recruited. Describe the screening process. Include a PRISMA diagram.

Describe the methods of fecal sample collection for each group. Were controls given home-sampling materials and how were samples stored and transported to the investigators?

Results: Include a table for participant characteristics.

Line 120: Do you mean to write the age range for adolescents instead of stating the number (20) twice?

Figures: Define the age groups in the footnotes.

Figure 4: Define the axis labels with units also.

Figure 5: Ratio of Firmicutes to ___?

Discussion: Describe strengths and limitations.

English editing is needed. Words are used inappropriately - I list a few examples.

Author Response

We are much appreciated to you for careful reading and useful comments!

The language needs editing. For example, these and other sentences/words need correction:

(1) Line 1. "sufficiently"
Answer 1. Changed to “over a wide range”.

(2) Line 20. "allowed to detect"
Answer 2. ‘allowed’ is changed to ‘enabled’.

(3) Abstract: the concluding sentence about new probiotics seems to have come out of no where. Perhaps conclude with the comparison of health to DM I microbiota populations.
Answer 3. Now we conclude the abstract with a sentence on probable replacement of Actinomycetota and Bifidobacter pool in T1DM.

(4) Methods: Include a statement about IRB approval of the protocol.
Answer 4. The IRB statement is included (lines 68-70).

(5) List inclusion and exclusion criteria and how participants were recruited. Describe the screening process. Include a PRISMA diagram.
Answer 5. The inclusion and exclusion criteria are listed in brief, both for the patients and control group (lines 78-87).

(6) Describe the methods of fecal sample collection for each group. Were controls given home-sampling materials and how were samples stored and transported to the investigators?
Answer 6The techniques of fecal sampling, storage and transportation are added (lines 89-95).

(7) Results: Include a table for participant characteristics.
Answer 7. Table 1 is on age and gender distribution in diabetes and controls is added.

(8) Line 120: Do you mean to write the age range for adolescents instead of stating the number (20) twice?
Answer 8. The misprint is corrected to '11 to 21 years old'

(9) Figures: Define the age groups in the footnotes.
Answer 9. We have added age ranges to Table 1.

(10) Figure 4: Define the axis labels with units also.
Answer 10. Relative contents of the two major phyla are now mentioned in the legend to Figure 4.

(11) Figure 5: Ratio of Firmicutes to ___?
Answer 11. Added: ‘… to the total bacterial mass’

(12) Discussion: Describe strengths and limitations.
Answer 12. Limitations of the study are mentioned (line 306)

(13) Comments on the Quality of English Language. English editing is needed. Words are used inappropriately - I list a few examples.
Answer 13. The text was carefully edited, and some inappropriate terms were replaced.

Round 2

Reviewer 2 Report

Line 29: IBD is inflammatory bowel dz.

84: Irritable bowel syndrome

92: sample should be immerse - needs correction

Tables 5-7: Be consistent in naming the healthy and control groups.

230: all subgroups NOT any

257: "obesity syndrome" is not a known term

Limitations belong in the Discussion

A PRISMA diagram should be included.

Some errors persist in the manuscript.

Author Response

Thank you for your useful and constructive comments!
Our replies are as follows:

(1) Line 29: IBD is inflammatory bowel dz.
Answer 1: Changed to inflammatory.

(2) 84: Irritable bowel syndrome
Answer 2: Corrected.

(3) 92: sample should be immerse - needs correction
Answer 3: Corrected.

(4) Tables 5-7: Be consistent in naming the healthy and control groups.
Answer 4: Corrected to 'control group' everywhere.

(5) 230: all subgroups NOT any
Answer 5: Corrected.

(6) 257: "obesity syndrome" is not a known term
Answer 6: corrected to just 'obesity'.

(7) Limitations belong in the Discussion
Answer 7: Added limitations at the beginning of the Discussion section (line ).

(8) A PRISMA diagram should be included.
Answer 8: The PRISMA diagram is commonly used to select records when preparing systematic reviews. Our study is not a systematic review. We have described the procedure for selecting T1DM patients and controls in an understandable way in the "Materials and methods" section.